# Analysis of the Candidate Genes and Underlying Molecular Mechanism of P198, an RNAi-Related Dwarf and Sterile Line

**DOI:** 10.3390/ijms25010174

**Published:** 2023-12-22

**Authors:** Shengbo Zhao, Junling Luo, Min Tang, Chi Zhang, Miaoying Song, Gang Wu, Xiaohong Yan

**Affiliations:** 1Key Laboratory of Biology and Genetic Improvement of Oil Crops, Ministry of Agriculture and Rural Affairs, Oil Crops Research Institute, Chinese Academy of Agricultural Sciences, Wuhan 430062, China; shengbo_ab@163.com (S.Z.); luojunling@caas.cn (J.L.); 15807171095@163.com (M.T.); anson-cheung@foxmail.com (C.Z.); songmy1027@163.com (M.S.); 2College of Life Science and Technology, Huazhong Agricultural University, Wuhan 430070, China; 3Key Laboratory of Agricultural Genetically Modified Organisms Traceability, Ministry of Agriculture and Rural Affairs, Oil Crops Research Institute, Chinese Academy of Agricultural Sciences, Wuhan 430062, China; 4Supervision and Test Center (Wuhan) for Plant Ecological Environment Safety, Ministry of Agriculture and Rural Affairs, Wuhan 430062, China

**Keywords:** RNAi, dwarfism, sterility, T-DNA, molecular characteristic, ONT sequencing, transcriptome

## Abstract

The genome-wide long hairpin RNA interference (lhRNAi) library is an important resource for plant gene function research. Molecularly characterizing lhRNAi mutant lines is crucial for identifying candidate genes associated with corresponding phenotypes. In this study, a dwarf and sterile line named P198 was screened from the *Brassica napus* (*B. napus*) RNAi library. Three different methods confirmed that eight copies of T-DNA are present in the P198 genome. However, only four insertion positions were identified in three chromosomes using fusion primer and nested integrated polymerase chain reaction. Therefore, the T-DNA insertion sites and copy number were further investigated using Oxford Nanopore Technologies (ONT) sequencing, and it was found that at least seven copies of T-DNA were inserted into three insertion sites. Based on the obtained T-DNA insertion sites and hairpin RNA (hpRNA) cassette sequences, three candidate genes related to the P198 phenotype were identified. Furthermore, the potential differentially expressed genes and pathways involved in the dwarfism and sterility phenotype of P198 were investigated by RNA-seq. These results demonstrate the advantage of applying ONT sequencing to investigate the molecular characteristics of transgenic lines and expand our understanding of the complex molecular mechanism of dwarfism and male sterility in *B. napus*.

## 1. Introduction

Forward genetics is an approach to determining the genetic basis of a specific phenotype. The genome-wide mutant library, constructed with reverse genetics methods such as Clustered regularly interspaced palindromic repeats (CRISPR)/Cas9 and RNA interference (RNAi), has accelerated the implementation of forward-genetic screening in plants [1,2,3,4]. Agrobacterium-mediated T-DNA transformation is widely used to generate these mutant libraries. Due to the random nature of this process, T-DNAs can be inserted into multiple sites across the entire genome. Insertion events occurring at a gene’s promoter or exon region disrupt its function [5,6,7,8]. In addition, complex T-DNA structures, including long T-DNA concatemers, incomplete T-DNAs, and vector backbone integration, are often present in transgenic plants [9,10,11,12,13]. Thus, obtaining the molecular characteristics of the transgenic line, including the T-DNA copy number, insertion locus, and sequences, is essential for identifying all potential genes related to the line’s phenotype.

The accurate evaluation of T-DNA copy numbers in transgenic lines is crucial for comprehensively identifying T-DNA insertion sites. Three different methods, namely, Southern blotting [14], quantitative polymerase chain reaction (qPCR) [15], and droplet digital PCR (ddPCR) [16], are well-known practical approaches to determining T-DNA copy numbers in transgenic lines. Southern blotting is considered a reliable method for accurately determining T-DNA copy numbers. However, this method is time-consuming, labor-intensive, and unsuitable for high-throughput analysis. In contrast, qPCR and ddPCR are fast approaches to evaluating the T-DNA copy number [15,16]. However, the precision of qPCR is affected by the amplification efficiency of the transgene elements and the endogenous reference gene. ddPCR estimates T-DNA copies by combining PCR sample partitioning, endpoint detection, and Poisson statistics [17]. Although ddPCR is more accurate, robust, and sensitive than qPCR in identifying T-DNA copies [16,18,19], the recombination of T-DNA in transgenic lines may affect its accuracy. Therefore, choosing the appropriate method to determine the T-DNA copy number facilitates the quick and accurate estimation of the T-DNA copy number.

Many methods based on PCR have been developed to obtain T-DNA insertion sites [20,21,22]. Among these methods, thermal asymmetric interleaved PCR (TAIL-PCR) is a practical approach to locating T-DNA insertion sites in transgenic plants, such as transgenic *Arabidopsis thaliana* [23,24], *Zea mays* [25], *Oryza sativa* (*O. sativa*) [26], *Nicotiana tabacum* [27], and *Populus* plants [28]. However, these PCR-based methods are time-consuming and cannot accurately identify all T-DNA insertion sites, especially those in exceptionally complex genomes [29,30]. As sequencing technology has advanced, next-generation sequencing (NGS) and single-molecule real-time (SMRT) sequencing have been successfully applied in transgenic plants to identify T-DNA insertion sites [31,32,33,34,35,36]. Unlike NGS, the long reads of SMRT sequencing allow the localization of T-DNA insertion sites and identification of inserted T-DNA sequences, especially in complex genomes [29,33].

Plant height and male sterility are critical agronomic traits used in production [37,38]. Dwarf plants and male sterile plants are widely utilized in breeding programs. Dwarf crops have shown improved lodging resistance, high resource efficiency, and improved yield index [39]. The utilization of the dwarfing gene *reduced height 1* (*Rht1*) in *Triticum* and *semi-dwarf1* (*sd1*) in *O.sativa* produced semi-dwarf varieties that greatly increased crop yields and triggered the “Green Revolution” in the 1960s [37,39,40]. Since then, many efforts have been made to identify new dwarf mutants to develop new varieties and to reveal the underlying molecular mechanism of dwarfism [41,42,43,44,45]. Although many dwarfing genes have been characterized in *Brassica napus* (*B. napus*), using these alleles in breeding is rarely reported [46,47,48,49,50]. Therefore, the identification of novel dwarf genes that are suitable for practical *B. napus* breeding is essential. Male sterility is also an important agronomic trait, which has been extensively exploited in the production of hybrid seeds for crop yield improvement through the harnessing of hybrid vigor [51,52,53,54,55]. Although many genes associated with male sterility have been identified, the mechanisms and gene networks during pollen development are still incomplete [53,56,57]. Therefore, identifying new male-sterile genes will deepen the understanding of plant fertility and facilitate the breeding of new varieties.

In this study, a dwarf and sterile line named P198 was screened from the *B. napus* long hairpin RNA interference (lhRNAi) library [4]. The molecular characteristics of P198 were investigated to identify potential genes underlying the P198 phenotype. Furthermore, transcriptomic analyses were conducted to identify the potential pathways and differentially expressed genes (DEGs) involved in the dwarfism and sterility phenotype of P198. These results expand our understanding of the complex molecular basis of dwarfism and male sterility in *B. napus*.

## 2. Results

### 2.1. Phenotypic Characteristics of the Dwarf and Male Sterile Line P198

The T0 generations of P198 and the wild-control Zhongshuang 6 (ZS6) were propagated in vitro by nodule explants to observe the P198 phenotype. P198 and ZS6 tissue culture seedlings of similar size were then grown in a greenhouse. At the seedling stage, mutant line P198 exhibited curled leaves with reduced length compared to the wild-type control ZS6 (Figure 1A,B). At the final flowering period, P198 showed an obvious dwarf phenotype (Figure 1C), with a height of 76.29 ± 10.20 cm (*n* = 7), which was 59.70% of the wild-type height (127.86 ± 11.38 cm, *n* = 7). In addition, P198 exhibited a male sterile phenotype with defective anthers (Figure 1D). P198 initially displayed well-developed stamens similar to those of ZS6 during the early stages of stamen development (marked with arrows in Figure 1E). However, alterations were then observed: the stamens of P198 were consistently smaller than those of ZS6 and eventually presented withered anthers with no visible pollen.

Paraffin sections of anthers from the tetrad to mature pollen stages were observed after staining with toluidine blue to determine the stage at which pollen development became impaired in P198. No differences were observed between P198 and ZS6 at the tetrad and uninucleate stages (Figure 2A–C,F–H). Differences between ZS6 and P198 occurred after the uninucleate stages. During the bicellular and mature pollen stages, the microspores of ZS6 underwent normal development and ultimately formed mature pollen grains (Figure 2D,E). In contrast, the cytoplasmic contents of P198 were greatly degenerated during the bicellular stage, and only slight inclusions were present in the anther during the mature pollen stage (Figure 2I,J). These findings indicate that microspore development is disrupted after the uninucleate stages.

### 2.2. Identification of Candidate Genes with T-DNA Insertion Mutations in P198

#### 2.2.1. Determination of T-DNA Copy Number in P198

T-DNA copy number was first determined to identify all T-DNA insertion mutation genes in P198. Southern blotting was initially performed with a digoxigenin-labeled *htpII* probe to determine the T-DNA copy number. The restriction enzymes *Bam*H I and *Hin*d III were chosen to digest the genomic DNA since each has a specific recognition site within the T-DNA (Appendix A). As shown in Figure 3, eight bands were obtained with *Bam*H I-digested genomic DNA of P198, whereas only five bands were detected when the genomic DNA of P198 was digested with *Hin*d III. The intense band in the *Hin*d III-digested P198 genome lane represents multiple fragments with similar sizes. Therefore, eight copies of exogenous T-DNA sequences are potentially present in P198.

qPCR and ddPCR were also conducted to estimate T-DNA copy numbers in the P198 genome. Specific primers and probes were designed based on the sequences of T-DNA and the endogenous reference gene *cruciferin A* (*CruA*) [58] (Appendix A). For qPCR, the standard curves of CruA, HPT II, NOS, and P35S were first analyzed. The correlation coefficients ranged from 0.998 to 1.000. The reaction efficiency of P35S, HPT II, and NOS (97.7%, 98.4%, and 101.8%, respectively) was similar to that of CruA (97.7%) (Appendix A). These results indicated that the qPCR systems used in this study were suitable for T-DNA copy number estimation. The T-DNA copy number was determined by comparing the absolute quantitative results of HPT II, NOS, or P35S elements with CruA. The corresponding estimated copy numbers were 7.87, 8.94, and 7.68 (Table 1). The T-DNA copy number was further analyzed using ddPCR. As shown in Table 1, 7.18, 6.93, and 7.08 copies of T-DNA were estimated with different primer sets. Although the estimated T-DNA copy numbers varied with different primer sets in qPCR and ddPCR, the results closely approximated that obtained by Southern blotting. These results suggested that there may be seven to nine exogenous sequence copies in P198.

#### 2.2.2. Detection of Insertion Sites with Fusion Primer and Nested Integrated PCR (FPNI-PCR)

Two sets of nested-specific primers targeting T-DNA sequences adjacent to the left or right border were designed (Appendix A). FPNI-PCR was conducted as previously described [59], and specific bands were obtained after three rounds of PCR amplification (Figure 4). The sequences of these specific bands were aligned to the ZS11 genome [60], and four insertion positions were identified in the P198 genome (Table 2). Two insertion positions, 609 bp apart, were identified in scaffoldA04. In contrast, only one insertion position was found in scaffoldC07 and scaffoldA07. As the obtained sequences only contain one end of the T-DNA sequences, it is unclear whether the two insertion sites in scaffoldA04 are derived from the same T-DNA insertion event, causing a 609 bp deletion in the chromosome, or from two different T-DNA insertion events. Therefore, it was postulated that the P198 line has three or four insertion sites.

#### 2.2.3. Detecting Copy Number and Insertion Sites with Oxford Nanopore Technology (ONT) Sequencing

Considering that the estimated T-DNA copy number was 7–9 but only 3–4 insertion sites were identified by FPNI-PCR, it was speculated that there might be unidentified T-DNA insertion sites or that tandem T-DNAs were present in P198. Therefore, P198 was resequenced using ONT to identify exogenous sequences and insertion sites throughout the genome. The statistics of the ONT sequencing for P198 are shown in Appendix A. A total of 38 Gb of data were generated, corresponding to a 33× sequencing depth, and 94.28% of the *B. napus* reference genome was covered by at least 1×. The average read length was 5500 bp, the N50 length was 2528 bp, and the mean read quality was 8.64.

All ONT reads containing the sequences of the transformation plasmid pMDC83 were extracted and de novo assembled using Caun software [61]. The obtained five contigs were then mapped to the transformed plasmid pMDC83 and *B. napus* ZS11 reference genome [60] to identify the T-DNA copy number and insertion sites. The alignment results of the contigs containing genome and T-DNA sequences are shown in Figure 5A. Both contig1 and contig5 carried a single copy of the T-DNA fragment. According to the physical positions of the contigs, these two T-DNA copies were integrated at position scaffoldA04:23593294–23593903 with a 609 bp genome deletion and scaffoldA07:30827957–30827977 with a 20 bp chromosome deletion. Contig2 and contig3 mapped to scaffoldC07 and contained only part of the T-DNA sequence. According to the junction point position of these two contigs on the reference genome, it was speculated that the exogenous DNA was inserted in scaffoldC07:58081673-58081721. Two representative ONT reads were further analyzed to explore the integrated T-DNA copy number in scaffoldC07. As shown in Appendix A, the exogenous sequences in read1 comprise three parts: intact T-DNA, partial T-DNA, and partial inverted T-DNA sequences. In contrast, read2 contains three copies of inverted T-DNA sequences. According to the direction of T-DNA in these two reads, we speculated that at least five copies of T-DNA were integrated in scaffoldC07. Therefore, at least seven copies of T-DNA were identified with ONT sequencing in three insertion sites. In addition, specific primers were designed based on T-DNA and its associated flanking sequences to confirm the reliability of the speculated T-DNA insertion sites (Appendix A). Gel electrophoresis of the PCR products demonstrated the expected bands (Figure 5B).

#### 2.2.4. Identification of T-DNA Insertion–Mutated Genes in P198

According to the insertion sites obtained above, the potential genes subject to insertional mutation in P198 were investigated (Appendix A). The T-DNA insertion sites in scaffoldA07 and scaffoldC07 were located in intergenic regions, and no functional gene was impaired by the T-DNA insertion. The T-DNA insertion site in scaffoldA04 were located in the long noncoding RNA (lncRNA) LOC111200331, and 609 bp spanning the first exon and intron was deleted (Figure 6). The expression of two transcript variants, X1 and X2, was interrupted. LncRNAs are regulators and play critical roles in various biological processes in plants [62]. Hence, *LOC111200331* may be an important candidate gene for the P198 phenotype.

### 2.3. Detection of Potential RNAi Target Genes

As P198 was screened from the *B. napus* lhRNAi library, the hairpin RNA (hpRNA) cassette could be present in the inserted T-DNA, which would induce sequence-specific gene silencing by degrading the target mRNA. Therefore, specific primers were designed to amplify the hpRNA construct (Appendix A). The sequences of specific bands (Figure 7A) were aligned against the ONT reads to identify the hpRNA cassette. Only one hpRNA cassette was identified in P198, and 119 bp hpRNA sequences in this cassette matched perfectly with the CDSs in LOC106353482 (*Brassica napus* protein cellulose synthase interactive 3 gene, *BnCSI3*) or LOC106353453 (*Brassica napus* protein cellulose synthase interactive 3-like gene, *BnCSI3L*). *BnCSI3* and *BnCSI3L* are homologous genes, and their exon sequences showed approximately 97.15% identity. The expression of *CSI3* and *CSI3L* (CSI3s) in shoots and buds was further analyzed by quantitative reverse transcription polymerase chain reaction (RT-qPCR). Compared with ZS6, the transcript levels of CSI3s did not change significantly in shoots but were reduced by approximately 30% in buds of P198 (Figure 7B), indicating that the hpRNA cassette in P198 was less effective.

### 2.4. Transcriptome Sequencing and DEG Analysis

mRNA sequencing (RNA-Seq) was performed to determine the underlying molecular mechanism related to the dwarf and sterile phenotype of P198. Three replicate shoots of P198 (P198-S1, P198-S2, P198-S3) and control ZS6 (ZS6-S1, ZS6-S2, ZS6-S3) were collected from explant cultured seedlings. Additionally, flower buds of P198 (P198-F1, P198-F2, P198-F3) and control ZS6 (ZS6-F-1, ZS6-F-2, ZS6-F-3) were collected. In total, 12 RNA sequencing libraries were prepared for transcriptome sequencing. After trimming adapters, low-quality bases, and contaminant reads, over 39 million clean reads with Q30 > 91% were obtained for each sample (Appendix A). The obtained clean reads were then aligned to the *B. napus* ZS11 reference genome; 85.87–92.93% of the clean reads uniquely mapped to the reference genome, indicating high-quality sequencing results. Pearson’s correlation coefficients among three replicates of various samples were ≥0.883 (Appendix A). Moreover, principal component analysis (PCA) showed that different samples could be clearly separated and that three biological replicates of these samples were closely clustered (Appendix A). Thus, the RNA-seq data were suitable for further analysis.

DEGs were identified from the RNA-seq data using the deseq2 R package. A total of 6682 DEGs (2678 upregulated and 4004 downregulated) were found in the shoots of P198 versus ZS6, and 4680 DEGs (1459 upregulated and 3221 downregulated) were found between P198 and ZS6 in buds. The expression of 10 randomly selected genes was investigated by qPCR to verify the reliability of the DEGs identified by RNA-seq. The identified expression patterns of these selected genes were similar to those obtained by RNA-seq (Figure 8A) and R^2^ = 0.87 (Figure 8B), indicating that the RNA-seq data are highly reliable.

### 2.5. GO and KEGG Enrichment Analyses

To investigate the biological functions of the DEGs, Gene Ontology (GO) term enrichment analysis was performed in three main GO categories, i.e., biological process (BP), molecular function (MF), and cellular component (CC). As shown in Figure 9A, the top overrepresented enriched GO terms among the DEGs in shoots mainly included the BP terms “auxin-activated signaling pathway”, “response to jasmonic acid”, “photosynthesis, light harvesting”, and “response to wounding”; the CC terms “photosystem II”, “cell wall”, “photosystem I”, and “plant-type cell wall”; and the MF terms “hydrolase activity, hydrolyzing O-glycosyl compounds”, “chlorophyll binding”, “heme binding”, and “FAD binding”. Moreover, the Kyoto Encyclopedia of Genes and Genomes (KEGG) enrichment analysis showed that “plant hormone signal transduction”, “photosynthesis-antenna proteins”, “alpha-linolenic acid metabolism”, “starch and sucrose metabolism”, and “phenylpropanoid biosynthesis” were the most enriched pathways in the DEGs of shoots (Figure 10A). These results indicate that the DEGs in shoots are mainly related to plant hormone signal transduction, photosynthesis, and cell wall biosynthesis.

The results of the GO analysis of the DEGs in the buds are shown in Figure 9B. The top enriched BP terms included “carbohydrate metabolic process”, “cell wall organization”, “pectin catabolic process”, and “cell wall modification”. The enriched CC terms included “extracellular region”, “apoplast”, “cell wall”, and “monolayer-surrounded lipid storage body”. The enriched MF terms included “transmembrane transporter activity”, “enzyme inhibitor activity”, “heme binding”, and “hydrolase activity, hydrolyzing O-glycosyl compounds”. The results of KEGG enrichment analysis showed that DEGs in buds were significantly enriched in “phenylpropanoid biosynthesis”, “ascorbate and aldarate metabolism”, “glycosaminoglycan degradation”, “alpha-linolenic acid metabolism”, “starch and sucrose metabolism”, and “galactose metabolism” (Figure 10B). These results indicate that the DEGs in buds are mainly associated with carbohydrate metabolism and cell wall biosynthesis.

### 2.6. Analysis of the DEGs in Shoots Involved in Phytohormone Biosynthesis and Single Transduction

Phytohormones are key regulators during plant growth and development. Compared with the wild-control ZS6, the P198 line displayed a dwarf phenotype. In addition, the GO term “auxin-activated signaling pathway” and the KEGG pathway “plant hormone signal transduction” were the most enriched terms among the DEGs. These results prompted us to further investigate the function of DEGs related to phytohormones, mainly auxin, in the P198 phenotype. After excluding genes with lower expression levels (FPKM < 5), 142 DEGs associated with phytohormone synthesis and signal transduction were identified (Appendix A). Among these DEGs, 61 were related to auxin biosynthesis and signal transduction (Appendix A). Regarding auxin biosynthesis, a gene (BnaC01G0100800ZS, *YUC8*) encoding a flavin monooxygenase-like protein was downregulated in P198, while two indole-3-acetaldehyde oxidase *AAO1* and three *NIT2* were upregulated. Regarding auxin transportation, one *polar auxin transport (PAT) 1* gene was upregulated, while one like aux1 (LAX) *2* gene, four *PIN3* genes, two *PIN7* genes, and four *Auxin1-resistant1 (AUX1)* genes were downregulated. Auxin/Indole-3-Acetic Acids (Aux/IAAs), Gretchen Hagen3s (GH3s), and small auxin upregulated RNAs (SAURs) have been identified as early and major auxin response genes [63]. Among these genes, 29 *IAA* genes were downregulated, five *GH3* genes showed altered expression levels in P198, and six *SAUR* genes were downregulated.

In the gibberellin (GA) biosynthesis and signaling pathway, 9 DEGs were identified in P198 versus ZS6 (Appendix A). Of these DEGs, two *GA20OX3* genes, which are involved in the biosynthesis of GA [64], were upregulated. One *GA2OX2* and two *GA2OX6* genes that participate in the metabolism of GA [65,66] were downregulated. The expression levels of two *GID1* genes (BnaC08G0417900ZS, *GID1B,* and BnaA06G0350800ZS, *GID1C*) were upregulated compared with those in ZS6. Two genes (BnaA09G0218400ZS, *RGA2* and BnaA10G0194400ZS, *RGL3*) encoding the DELLA protein were downregulated.

In contrast to ZS6, 12 BR-related genes were differentially expressed in P198 (Appendix A). In the BR synthesis pathways, the expression of a gene (BnaC07G0322400ZS, *CYP85A2*) encoding CYP85A2, which possesses BR C-6 oxidase and BL synthase activity [67], was significantly downregulated. Additionally, 11 genes related to BR signaling were differentially expressed in P198.

Furthermore, several genes involved in the biosynthesis and signal transduction pathways of cytokinin (CK), abscisic acid (ABA), and ethylene (ETH) were differentially expressed between P198 and wild-type ZS6 (Appendix A). These modified genes may interact with IAA, GA, and BR transcription factors to regulate P198 growth.

### 2.7. Analysis of Potential Key DEGs in Buds Associated with Male Sterility

Anther development has been well studied in the model plant *Arabidopsis thaliana*. Therefore, the potential DEGs in buds associated with male sterility were selected based on functional annotation with the Arabidopsis Information Resource (TAIR) database. A total of 31 candidate DEGs involved in anther development processes were identified, such as lipid metabolism, transporter, microspore division, and sugar metabolism (Appendix A). Among these genes, *ABCG9*, *ABCG16*, *ABCG31*, and *LHT2* are involved in transporting nutrient substances to supply pollen development [68,69,70]. *UGE3* and *UGP1* participate in sugar *metabolism* during pollen development [71,72]. *GPT1* plays a crucial role in transporting glucose 6-phosphate into nongreen plastids and is essential for gametophyte development [73]. *MS2*, *PGDH1*, *GPAT1*, and *GPAT6* are essential for lipid metabolism in pollen [74,75,76,77]. *PIRL1* and *PIRL9* are needed for microspore mitosis.

## 3. Discussion

### 3.1. LOC111200331 Is an Important Candidate Gene for the P198 Phenotype

Obtaining the molecular characteristics of transgenic lines, including T-DNA copies, insertion sites, and sequences, is critical to identifying the candidate genes related to the mutant phenotype. As transformants with multiple T-DNA insertions are frequent occurrences after *Agrobacterium tumefaciens*-mediated transformation [78,79,80], accurately determining the inserted T-DNA copies is crucial to comprehensively identify T-DNA insertion sites. Southern blotting, qPCR, and ddPCR have been widely used to measure transgene copy number. Despite being time-consuming and labor-intensive, Southern blotting is considered the most unambiguous method for estimating T-DNA copy numbers. In this study, the Southern blotting results varied with different restriction enzymes, yielding a maximum of eight bands, indicating that eight copies of T-DNA may be present in the P198 genome. qPCR is a faster and more sensitive approach to evaluating T-DNA copies [15,81,82]. However, qPCR results are confounded by amplification efficiency differences [15,83]. Therefore, identical reaction efficiencies of the transgene elements and the endogenous reference gene [15,84] are required to obtain accurate results. In this study, the reaction efficiencies of the transgene elements and the endogenous reference gene were similar. The estimated number of T-DNA copies (7.86–8.96) was consistent with that revealed by Southern blotting (eight copies). These results suggest that the established qPCR method for determining T-DNA copy number was reliable in this study. Therefore, it can be concluded that eight copies of T-DNA are present in the P198 genome. Although the ddPCR method is more robust and accurate than qPCR [16,18,19], ddPCR underestimated the T-DNA copy number in this study. Whether this effect was caused by the recombination of T-DNA into exogenous sequences or other unknown factors needs to be further studied.

Although many PCR-based methods have been successfully applied to locate T-DNA insertion sites in transgenic lines [20,21,22], it remains challenging to identify all insertion sites in multiple T-DNA-inserted lines using these methods, especially in complex genomes [29,30]. In this study, the T-DNA insertion sites were first investigated with FPNI-PCR, and only four T-DNA insertion positions in three chromosomes were identified. Only one end of the flanking sequences for two insertion sites was obtained, potentially due to the complex flanking sequences of insertion sites. SMRT sequencing is a more efficient approach than PCR-based methods for identifying T-DNA flanking sequences and insertion sites [29,33]. In this study, three T-DNA insertion sites and the corresponding flanking sequences at both ends were obtained. In addition to single-copy T-DNA insertion, T-DNA can also be integrated as long T-DNA concatemers [10]. In this study, one copy of T-DNA was inserted into scaffoldA07 and scaffoldA04. In addition, a T-DNA concatemer containing at least five copies of T-DNA was identified in scaffoldC07. Therefore, at least seven copies of T-DNA were found in P198 by ONT sequencing, lower than the estimated T-DNA copy number. This finding may result from the relatively low ONT sequencing depth of P198 and the repeated T-DNA sequences presented in scaffoldC07, which are longer than the maximum length of ONT reads.

Based on the hpRNA cassette sequences and T-DNA insertion sites, three potential mutant genes, namely, *BnCSI3*, *BnCSI3L*, and *LOC111200331*, were identified in this study. *BnCSI3* and *BnCSI3L* are predicted RNAi target genes. Both genes are homologous to the *Arabidopsis thaliana* AtCSI3 gene, which is involved in cellulose synthesis and cell elongation. The mutation of *AtCSI3* alone does not result in any observable phenotype; however, it can enhance the reduced hypocotyl length phenotype of *AtCSI1* mutants [85]. In this study, the transcript levels of CSI3s were downregulated by approximately 30% in P198 versus ZS6 buds, and no differential expression was observed between P198 and ZS6 shoots. These results indicate that the hpRNA cassette in P198 was not efficient enough to induce CSI3 silencing in plants. Therefore, CSI3s may not be candidate genes related to the P198 phenotype, and the downregulated CSI3s in buds may be caused by other factors that need to be further investigated. LncRNAs widely influence plant growth and development [62]. However, the conservation of lncRNA sequences among different species is low, and no homologous lncRNAs of *LOC111200331* were found in other species. Whether the mutation of *LOC111200331* contributes to the phenotype of P198 needs to be further studied.

### 3.2. Auxin and BR Are Critical for Curled Leaves and the Dwarf Phenotype in P198

Phytohormones, particularly auxin, GA, and BR, are vital regulators of plant growth and development [86]. Auxin acts as a key regulator at all stages of plant growth. Numerous mutations altering the synthesis, transportation, and response of auxin result in dwarfism, vascular defects, and abnormal leaf morphology [87,88]. The YUC pathway has been considered a primary biosynthetic route for IAA. In this study, a *YUCCA8* gene encoding a flavin monooxygenase-like protein that catalyzes a rate-limiting step in the YUC pathway [89] was found to be downregulated 4.88-fold in P198. The reduced expression of this gene may lead to reduced IAA synthesis in P198, thus affecting P198 height. Modifying auxin activity through PAT is critical for plant development. AUX1/LAX and PINs play a crucial role in PAT [90]. In this study, one *PIN1* gene was upregulated, while one *LAX2* gene, four *PIN3* genes, two *PIN7* genes, and four *AUX1* genes were downregulated. The altered expression of these auxin transporters may affect auxin distribution and influence the growth of P198.

*AUX/IAA*, *GH3*, and *SAUR* have been recognized as major early auxin response genes [63]. Auxin-mediated transcriptional regulation is mainly dependent on the function of Aux/IAAs [91]. Mutations in Aux/IAA proteins can block auxin signal transduction and lead to defects in various developmental processes [92,93,94]. For example, a mutation in the *IAA2* gene of *B. napus* results in semidwarfism and curly leaves [95]. Mutations in the *IAA7* genes of *B. napus* affect stem elongation, reproduction, and plant height [96,97]. In this study, eight *IAA2* genes, three *IAA3* genes, three *IAA7* genes, one *IAA8* gene, two *IAA9* genes, one *IAA12* gene, two *IAA13* genes, one *IAA16* gene, three *IAA17* genes, one *IAA19* gene, one *IAA28* gene, and three *IAA29* genes were significantly downregulated in P198. The downregulated expression of these *IAA* genes hampers auxin signal transduction, leading to curled leaves and a dwarf phenotype in P198. GH3 proteins act as negative regulators in response to auxin. In addition, GH3.3, GH3.5, and GH3.6 can adjust jasmonic acid homeostasis by mediating its conjugation to amino acids [98]. Compared with ZS6, in P198, one *GH3.3* gene was upregulated, while one *GH3.10* gene, one *GH3.6* gene, and two *GH3.5* genes were downregulated. The altered expression of these GH3 genes indicates that IAA and JA may function coordinately to affect the growth of P198.

GA is a crucial phytohormone in determining plant height. In this study, two *GA20OX3* genes associated with GA synthesis [64] were upregulated. However, one *GA2OX2* and two *GA2OX6* genes involved in GA metabolism [65,66] were downregulated. In the GA regulatory pathway, two *GID1* genes (*GID1B* and *GID1C*) were upregulated, and two genes encoding the DELLA protein were downregulated in P198 compared with those in the control. The upregulated GID1 and downregulated DELLA genes are expected to relieve DELLA-mediated growth inhibition [99] and promote the growth of P198. However, P198 was shorter than the control. These results suggest that GA has no effect on the height of P198 and that the altered expression of GA-related genes may be caused by other factors.

In the synthesis pathways of BR, CYP85A2 is an enzyme that exhibits bifunctional characteristics by possessing BR C-6 oxidase and BL synthase activity. The mutation of the *CYP85A2* gene in Arabidopsis resulted in dark green and curled leaves [67]. In this study, curled leaves were observed; thus, this gene may be involved in the P198 phenotype. As a primary receptor of BR, BRI1 is essential for cell elongation and vascular development [100,101]. In this study, two BRI1 genes were significantly downregulated, indicating that BRI1 may play a role in the P198 dwarf phenotype.

### 3.3. Alterations in the Expression of Genes Associated with Lipid Metabolism and the Tapetum Secretion Process May Lead to Abnormal Pollen in P198

During pollen development, the tapetum layer cells synthesize and secrete diverse substances, including lipids, to facilitate microspore development [102,103]. Thus, disrupting the synthesis and secretion pathway in the tapetum layer cells could affect the normal development of microspores and lead to male sterility. Lipid metabolism plays a vital role in anther development [104,105]. Among the lipid synthesis pathways, *GPAT1* and *GPAT6* play important roles in tapetum and pollen development [74,75]. *PGDH1* encodes a 3-phosphoglycerate dehydrogenase required for tapetum differentiation and male fertility [76]. *MS2* encodes a fatty acid reductase essential for pollen wall development [77]. In this study, two *GPAT1*, four *GPAT6*, two *MS2*, and two *PGDH1* genes were downregulated. These results suggest that lipid synthesis is altered in P198, which may alter the lipid supplement of microspores and lead to the male sterility phenotype of P198.

Many transporters participate in the secretion process of material from tapetal cells to anther locules for developing microspores [106,107]. ABCG9 and ABCG31 are two plasma membrane ABC transporters that transfer steryl glycosides for trypsin deposition [68]. ABCG16 is involved in transporting lipid precursors and polysaccharides required for nexin and intine formation [69]. AtLHT2 is crucial for transporting neutral and acidic amino acids into tapetum cells to synthesize substances essential for microspore structure and exporting organic nitrogen into the locule for pollen development [70]. In this study, one *ABCG9*, two *ABCG16*, four *ABCG31*, and two *LHT2* genes were downregulated. These downregulated genes may reduce the associated nutrient supply of the microspore and lead to male sterility.

## 4. Materials and Methods

### 4.1. Plant Materials

The P198 line was obtained from the lhRNAi library of *B. napus*. P198 and the control ZS6 were both propagated in vitro to quickly obtain seedlings for further analysis. Briefly, explants were excised from lateral branches with only one node and then sterilized by immersion in 0.1% sodium hypochlorite solution for 10 min, followed by rinsing five times with sterile distilled water. The sterilized explants were then transferred onto M4 media [108] to induce root formation. Explant-cultured seedlings of similar size from P198 and ZS6 were selected to be planted in the greenhouse (22 °C, 16 h of light per day) for the subsequent studies.

### 4.2. Southern Blotting

The DNA of P198 and the wild-control ZS6 were extracted using a modified CTAB method [109]. The purified DNA was then used for the following study. Thirty micrograms of genomic DNA from P198 and ZS6 were digested with *Bam*H I and *Hin*d III and then separated on a 0.8% agarose gel. One microgram of the positive control plasmid pMDC83 was treated under the same conditions. The alkali-resolved DNA was then transferred to a Zeta-Probe membrane (Bio-Rad, Hercules, CA, USA, Cat. No. 1620159) and fixed by UV crosslinking. The DIG-labelled *hptII* probe was prepared with the DIG PCR Probe Synthesis Kit (Roche, Mannheim, Germany, Cat. No. 11636090910). Following hybridization, the membrane was visualized by autoradiogram Tanon 1600 (Tanon, Shanghai, China).

### 4.3. qPCR

qPCR was performed to estimate the T-DNA copy number as previously described [15]. Specific primers and probes for T-DNA elements and the reference gene *CruA* were designed (Appendix A). The plasmid pMDC83-CruA containing qPCR detection sequences for the *CruA* gene was constructed. pMDC83-CruA was first diluted to 2.62 × 10^8^ copies/µL with TE buffer. The solution was then 8-fold serially diluted to final concentrations of 3.28 × 10^7^, 4.10 × 10^6^, 5.12 × 10^5^, 6.40 × 10^4^, and 8.00 × 10^3^ copies/µL. The serially diluted solutions were used as standard solutions to generate the standard curves. The genome of P198 was digested with *Hin*d III and *Bst* XI, and 2 µL of digested DNA was used as the template for qPCR.

Reactions were conducted using Premix Ex Taq™ (Probe qPCR) (Takara, Tokyo, Japan, Cat. No. RR390A) in a final volume of 10 µL containing 1 × Conc. Premix Ex Taq (Probe qPCR), 400 nmol forward and reverse primers, and a 200 nmol probe. All reactions were performed in the CFX384 Touch Real-Time PCR Detection System (Bio-Rad, Hercules, CA, USA) under the following conditions: 95 °C for 1 min, followed by 40 cycles of 10 s at 95 °C and 1 min at 60 °C (acquisition of fluorescence signal). The concentrations of P35S, NOS, HPT II, and CruA were determined by interpolation of their PCR signal (Cq) into the corresponding standard curve. T-DNA copy number was determined by comparing the concentration of P35S, NOS, or HPT II elements with the concentration of the *CruA* gene. Three technical replicates were conducted for each primer set. The standard curves and no-template control were included in all runs.

### 4.4. ddPCR

Primers and genomic DNA were the same as those used for qPCR. The ddPCR assay was performed on the QX200™ droplet digital™ PCR system (Bio-Rad, Hercules, CA, USA) according to the manufacturer’s instructions. Briefly, 20 μL of ddPCR mixtures containing 1× ddPCR supermix (Bio-Rad, Hercules, CA, USA, Cat. No. 1863010), 800 nmol of each primer, 400 nmol of probe, and 1 μL of template were used to generate droplets. The emulsified samples were then transferred to PCR plates and amplified in a C1000 Touch™ deep-well thermal cycler (Bio-Rad, Hercules, CA, USA). After amplification, the droplets of each sample were analyzed in the QX200 droplet reader (Bio-Rad, Hercules, CA, USA), and the concentrations were analyzed using QuantaSoft software (version 1.7.4, Bio-Rad, Hercules, CA, USA). The T-DNA copy number was determined by comparing the concentration of P35S, NOS, or HPT II elements with the concentration of the *CruA* gene. Four technical replicates were performed per reaction.

### 4.5. FPNI-PCR

FPNI-PCR was conducted to identify the T-DNA insertion sites as previously described [59]. The fusion primers used in this study were identical to those previously described, and the T-DNA-specific primers were designed based on sequences close to the left border (LB) or right border (RB) of the T-DNA (Appendix A). The 2× Phanta Max Master Mix (Vazyme Biotech Co., Nanjing, China, Cat. No. P515-02) with high-fidelity enzyme was used for all reactions. All PCR conditions used in this study were identical to those previously described. Sanger sequencing was performed on the obtained specific bands using the FSP2 primer. The sequences of these bands were then mapped to the ZS11 reference genome to identify T-DNA insertion loci.

### 4.6. ONT Sequencing

The P198 genome was resequenced using the ONT platform to identify the insertion sites. After filtering out low-quality reads and adapter sequences, all clean reads were mapped to pMDC83 sequences using minimap2 (version 2.20) [110]. The mapped reads were extracted using samtools (version 1.13) [111] and converted to the fastq file format, which was then used for de novo assembly with canu (version 2.2) [61]. The assembled contigs were then aligned to the ZS11 genome and pMDC83 sequences to identify T-DNA insertion sites.

### 4.7. Transcriptome Analysis

Shoots of P198 and ZS6 were collected from explant-cultured seedlings; seedlings of a similar size were used in RNA-seq to minimize the effect of environmental factors on the transcriptome analysis. The buds of P198 and ZS6 were also collected from the same plants. Total RNA was extracted with TRIzol reagent (Life Technologies, Waltham, MA USA, Cat. No. 15596026) and subsequently used to construct RNA libraries. These libraries were sequenced on the Illumina NovaSeq 6000 sequencing platform (Illumina Inc., San Diego, CA, USA) in PE150 mode. After trimming adapters, low-quality bases, and contaminant reads, the clean reads were aligned to the ZS11 reference genome using HISAT2 (version 2.0.4) [112]. The count number of each gene was then calculated. DEGs between P198 and ZS6 were identified using the R package DESeq2 (version 1.30.1) [113] with parameters FDR < 0.01 and |FoldChange| ≥ 3. GO and KEGG enrichment analyses of the DEGs were then performed using the R package ClusterProfiler (version 4.4.4) [114].

### 4.8. RT–PCR

Total RNA was extracted from three independent replicates of buds and shoots. The obtained RNA was subsequently used to synthesize cDNA using the HiScript III 1st Strand cDNA Synthesis Kit (+gDNA wiper) (Vazyme Biotech Co., Nanjing, China, Cat. No. R312-02). qPCR was performed using realuniversal color premix (SYBR green) (Tiangen Biotech Co., Beijing, China, Cat. No. FP201-02) on the CFX384 touch real-time PCR detection system (Bio-Rad, Hercules, CA, USA). All reactions were performed in a 10 µL volume with the recommended reaction system. *Bnactin7* was used as the internal reference gene, and all primers used in this study are listed in Appendix A. The expression level of each gene was calculated using the 2^−△△^Ct method [115].

## Figures and Tables

**Figure 1 ijms-25-00174-f001:**
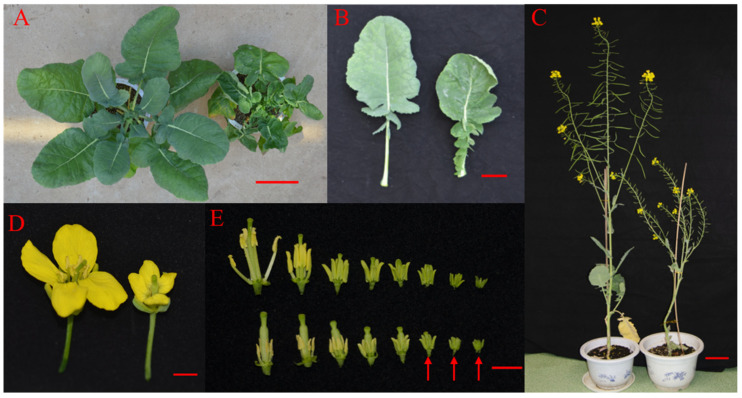
Phenotypes of P198. (**A**,**B**) Plants (**A**) and leaves (**B**) of ZS6 (left) and P198 (right) at the seeding stage. (**C**) Plant heights of ZS6 (left) and P198 (right) at the final flowering period. (**D**) Flower of ZS6 (left) and P198 (right). (**E**) Comparison of stamens from ZS6 (top) and P198 (bottom) in different developmental stages. The arrows indicate well-developed stamens of P198 similar to those of ZS6. Scale bars: 10 cm in (**A**,**C**), 5 cm in B, and 1 cm in (**D**,**E**).

**Figure 2 ijms-25-00174-f002:**
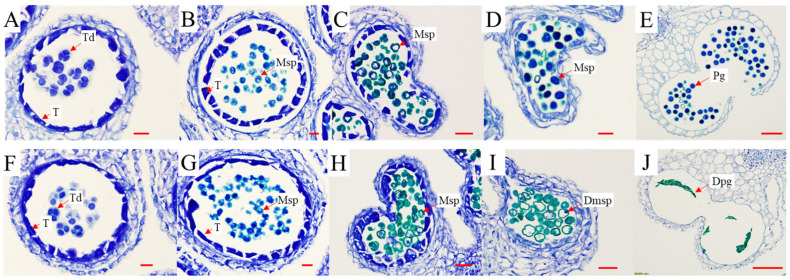
Comparison of transverse anther sections from wild control ZS6 (**A**–**E**) and transgenic line P198 (**F**–**J**). All sections of the anther were stained with toluidine blue. Microspores at the tetrad (**A**,**F**), early uninucleate (**B**,**G**), late uninucleate (**C**,**H**), bicellular (**D**,**I**), and mature pollen (**E**,**J**) stages were observed. T, tapetum; Td, tetrad; Msp, microspore; Pg, pollen grain; Dmsp, degraded microspore; and Dpg, degraded pollen grain. Scale bars: 10 μm in (**A**,**B**,**F**,**G**), 20 μm in (**C**,**D**,**H**,**I**), and 50 μm in (**E**,**J**).

**Figure 3 ijms-25-00174-f003:**
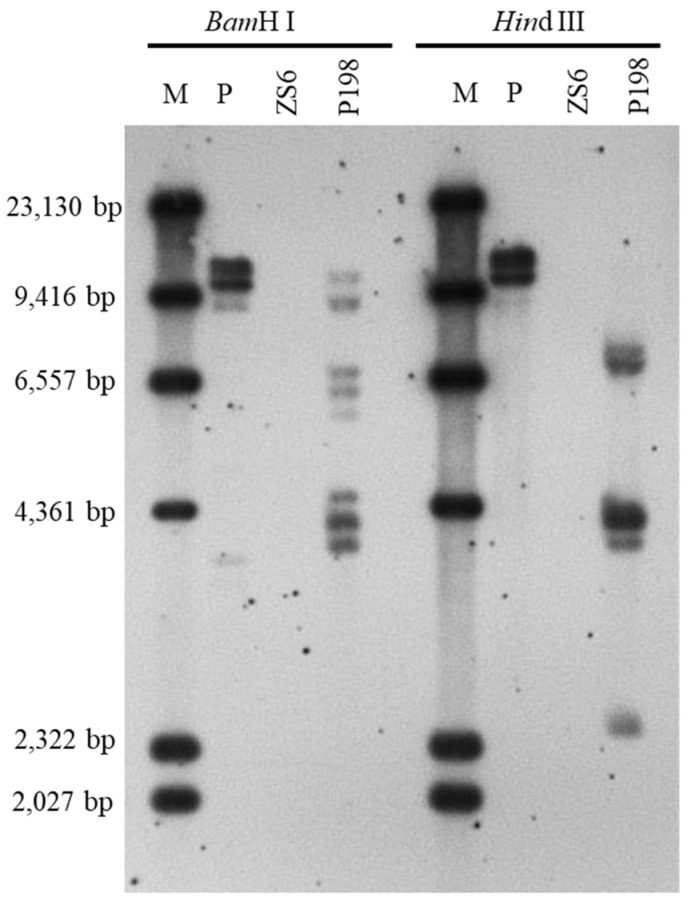
Southern blotting of transgenic line P198. DNA was digested with *Bam*H I or *Hin*d III and subjected to hybridization with the digoxigenin-labeled *hptII*-specific probe. P, positive control, pMDC83 plasmid DNA; ZS6, negative control, wild-type line ZS6.

**Figure 4 ijms-25-00174-f004:**
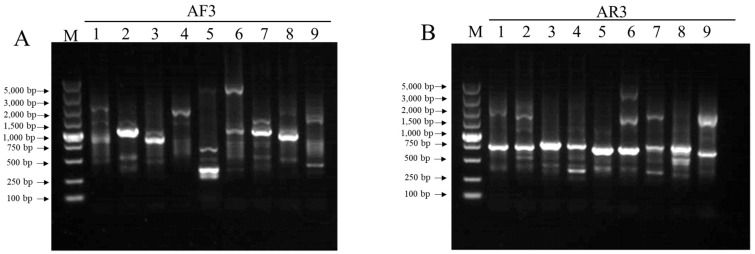
Identification of T-DNA flanking sequences with FPNI-PCR. (**A**) PCR amplification products were generated with three specific primers (AF1–AF3) and 1–9 fusion arbitrary degenerate primers (FPs). (**B**) PCR amplification products were generated with three specific primers (AR1–AR3) and 1–9 FPs. M, DL5000 DNA marker.

**Figure 5 ijms-25-00174-f005:**
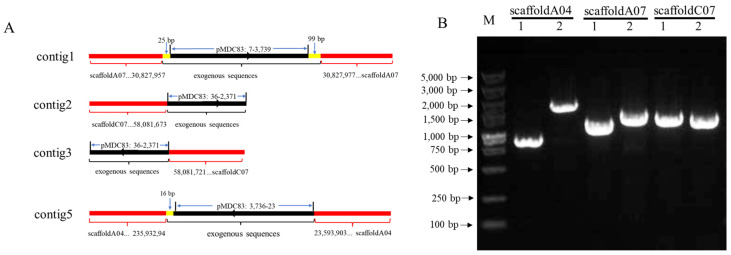
Insertion analysis of P198 using ONT sequencing. (**A**) The sketch map of contig alignment to pMDC83 and the reference genome. The yellow fragments indicate unknown sequences, and the red fragments indicate the identified T-DNA flanking sequences. The arrows indicate the T-DNA direction from LB to RB. (**B**) PCR verification of both ends of T-DNA insertion sites in P198; 1 and 2 indicate primers that amplified the 5′ and 3′ junction sequences in the P198 line, respectively. M, DNA marker DL5000.

**Figure 6 ijms-25-00174-f006:**
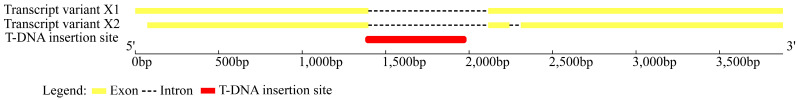
Schematic representation of the T-DNA insertion–mutated gene *LOC111200331* in P198. The insertion site spanned the first exon and intron, and the expression of two transcript variants, X1 and X2, was interrupted.

**Figure 7 ijms-25-00174-f007:**
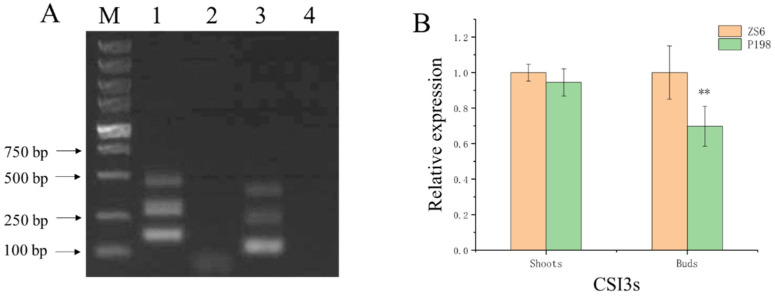
Analysis of RNAi target genes. (**A**) PCR amplification of the hpRNA construct in P198. Lanes 1–2 and 3–4 show the products of primer sets RNAi-F and RNAi-R amplified in P198 and the negative control (water), respectively. (**B**) Expression level detection of CSI3s by RT-qPCR. ** indicate a statistically significant difference compared to ZS6 (*p* < 0.05).

**Figure 8 ijms-25-00174-f008:**
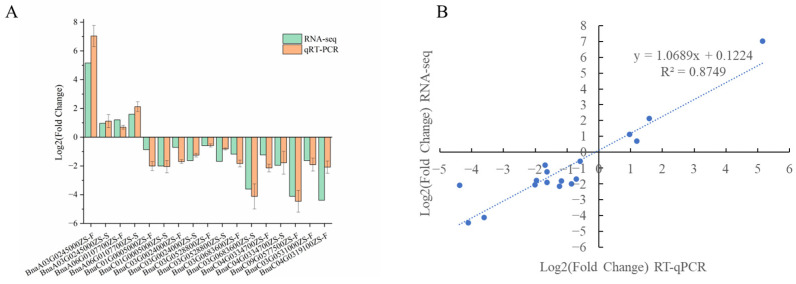
Validation of RNA-seq results by qPCR. (**A**) Log 2-fold changes of randomly selected DEGs. S, shoots; F, buds. (**B**) The correlation of gene expression patterns determined by qPCR (x-axis) and RNA-seq (y-axis).

**Figure 9 ijms-25-00174-f009:**
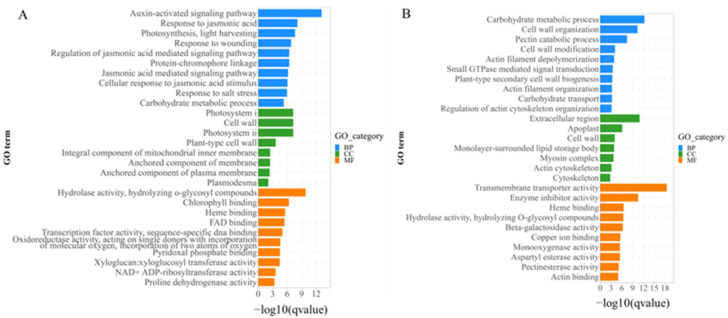
Significantly enriched GO terms of the DEGs between P198 and wild-type ZS6. (**A**) The top 10 significantly enriched BP, MF, and CC terms of the DEGs with *p* < 0.01 in shoots. (**B**) The top 10 significantly enriched BP, MF, and CC terms of the DEGs with *p* < 0.01 in buds.

**Figure 10 ijms-25-00174-f010:**
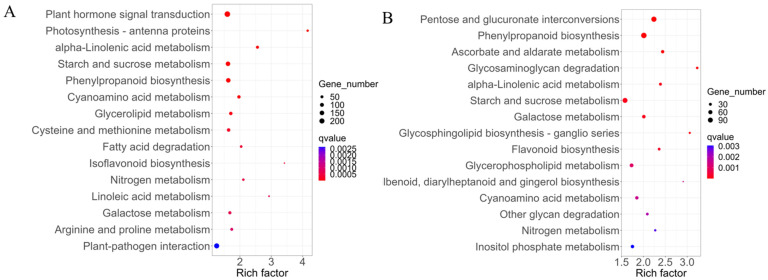
Top 15 significantly enriched KEGG pathways of the DEGs in shoots (**A**) and buds (**B**) between P198 and wild-type ZS6. The enrichment factor is the ratio of the total gene number to the DEG number in a certain pathway. The point size indicates the number of DEGs corresponding to the pathway, and the color represents the q-value.

**Table 1 ijms-25-00174-t001:** Estimation of T-DNA copy numbers by qPCR and ddPCR using primer sets targeting the *CruA* reference gene or different T-DNA elements (±se; *n* = 3 for qPCR; *n* = 4 for ddPCR).

Method	Primer Set Name	Concentration (Copy/μL)	Ratio to *CruA*	T-DNA Copy Number
qPCR	CruA	9,057,666.67 ± 549,797.54	1.00 ± 0.05	-
	HTP II	17,803,333.33 ± 1,201,013.46	1.97 ± 0.08	7.87 ± 0.3
	NOS	20,300,000.00 ± 2,714,479.69	2.23 ± 0.14	8.94 ± 0.56
	P35S	17,373,333.33 ± 489,114.85	1.92 ± 0.05	7.68 ± 0.21
ddPCR	CruA	47.98 ± 1.69	1.00 ± 0.04	-
	HTP II	172.00 ± 1.22	3.59 ± 0.13	7.18 ± 0.27
	NOS	82.95 ± 1.69	1.73 ± 0.09	6.93 ± 0.36
	P35S	84.73 ± 2.51	1.77 ± 0.11	7.08 ± 0.43

**Table 2 ijms-25-00174-t002:** T-DNA insertion sites identified using FPNI-PCR.

Chromosome	Insertion Sites	Adjacent T-DNA Board	Strand
scaffoldA04	23,593,294	RB	+
scaffoldA04	23,593,903	LB	−
scaffoldC07	58,081,718	LB	−
scaffoldA07	30,827,977	RB	−

Strand indicates the strand of the genome to which the sequences of specific bands are mapped.

## Data Availability

The data presented in this study are available in this article and its Appendix A.

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
