# Peer review of "Analysis of the Candidate Genes and Underlying Molecular Mechanism of P198, an RNAi-Related Dwarf and Sterile Line"

_ijms, 2023, doi:10.3390/ijms25010174_

Round 1

Reviewer 1 Report

Comments and Suggestions for Authors

This manuscript is informative and offers accurate experiments and excellent analysis and discussion. However, a few minor corrections are needed and these have been noted directly in the manuscript. Please refer to the attached manuscript file. 

Author Response

Thank you very much for taking the time to review this manuscript. Please find detailed responses in the attachment, and corresponding revisions have been marked in red in the resubmitted manuscript.

Reviewer 2 Report

Comments and Suggestions for Authors

The purpose of this work was to characterize the genetics of a Brassica napus lnRNAi line with dwarfism and male sterility, two striking phenotypes with agricultural and research interest.  The authors used Southern blotting, qPCR, and sequencing to characterize the number, location, and arrangement of what turned out to be multiple insertions in the genome.  They also performed expression analysis on genes in shoots and buds of both WT and the lnRNAi line to try and determine the reason for the altered phenotype.  This paper was an interesting read and used alternative approaches to puzzle out the number of inserted T-DNA copies.  The discussion of the benefits and limitations of each approach was well done. 

Main concerns

1.       Please add a citation for where the lnRNAi plant was obtained from.  This is a small but key detail. 

2.       The phenotyping of the plant line of interest needs to be quantified.  The pollen development was well-tested and presented.  However, the leaf traits and overall plant height need quantification.  Are the lines impaired in female sterility as well?  The representative image shows unexpanded seed pods, was this due to growth in isolation from viable pollen, or due to a lack of female fertility as well?  If the line is female fertile then perhaps breeding could be used to segregate out the insertions in different contigs from each other (assuming lack of tight linkage between contigs) and aid in parsing out which insertion is key for the observed phenotype.  

3.       It is unclear if the T-DNA line with multiple insertions is a standard outcome for this particular library or a relatively frequent occurrence.  How common are multiple T-DNA insertions in the B napus lhRNAi library?  How does this frequency compare with other T-DNA collections?  

4.       It would be very helpful to show a diagram of the T-DNA structure with features (NOS etc), restriction sites, and overall size.  This information may aid the reader in interpreting the results from the various T-DNA detection and quantification approaches.  On that note, it appear there is one rather dark band for the HindIII digest, might this band represent multiple fragments of a similar size?  This lane gave the lowest estimate for the insertion number but is perhaps an underestimate. 

5.       The importance of the height and sterility traits in the introduction ought to be expanded upon.  This topic could include green-revolution semi-dwarf crops, the use of smaller cultivars of fruit trees etc., along with the utility of male-sterile lines in generating hybrid corn. 

6.       The gene expression analysis and annotation thereof as currently presented really doesn’t add much to the discovery of why the particular plant has the dwarf and sterile phenotype.  I would rather have seen more focus on the lnRNAi aspect of the work.  For example, attempting to identify targets of the disrupted LOC11120031 via BLAST or similar approaches.  Did the remaining T-DNAs not contain a hairpin cassette?  Please comment. 

Minor concerns

7.       Please add full genus species names before all common names (such as in line 65), italicize all genus species names. 

8.       In Figure 1, please reverse the leaf image so that WT is on the left and mutant on the right as a match to the other panels. 

9.       Please move figure S1 from main to supplementary, same for S2, and S3 (unless this is standard for the journal at this stage, in which case it will be changed during later article steps). 

10.   Please do not use red and green and informative colors in figures, not all readers can see the difference.  Please use a different color/pattern for parts of Figure 5, Figure S2, Figure S3, and Figure 10. 

11.   The lettering on figure 2 is very difficult to read, please use a different font color and/or imbed the letters in a textbox with a white background. 

12.   Please increase the size of figures 8, 9, and 10.  At the current size the font is quite small. 

Comments on the Quality of English Language

No concerns. 

Round 2

Reviewer 2 Report

Comments and Suggestions for Authors

Thank you for the extensive revisions to your manuscript.  I agree that the leaf traits are challenging to quantify overall, here the general description works well, and the picture of the overall rosette is a very nice addition.  

I would suggest briefly adding a mention of the female sterility trait as well, just in case the causative gene is later definitively characterized/identified.  Perhaps the lncRNA is important for meiosis.